# Maternal, paternal, and other caregivers' stimulation in low- and- middle-income countries

Jorge Cuartas[1]*, Joshua Jeong[2], Catalina Rey-Guerra[3], Dana Charles McCoy[1], Hirokazu Yoshikawa[4]

1 Harvard Graduate School of Education, Harvard University, Cambridge, Massachusetts, United States of America, 2 T.H. Chan School of Public Health, Harvard University, Cambridge, Massachusetts, United States of America, 3 Lynch School of Education and Human Development, Boston College, Boston, Massachusetts, United States of America, 4 Steinhardt, New York University and Global TIES for Children, New York City, New York, United States of America

* jcuartas@g.harvard.edu

## Abstract

### Background and objectives

Globally, studies have shown associations between maternal stimulation and early child development. Yet, little is known about the prevalence of paternal and other caregivers' stimulation practices, particularly in low- and- middle-income countries (LMICs).

### Methods

Data from the Multiple Indicators Cluster Survey (MICS) and the Demographic and Health Survey (DHS) were combined across 62 LMICs (2010–2018). The sample included 205,150 mothers of children aged 3 and 4 years. High levels of stimulation were defined as caregiver engagement in at least 4 out of 6 possible activities with the child. The proportion of mothers, fathers, and other caregivers providing high levels of stimulation was calculated by country, region, and for the whole sample. Socioeconomic disparities within and between countries were estimated.

### Results

On average, 39.8% (95% CI 37.4 to 42.2) of mothers, 11.9% (95% CI 10.1 to 13.8) of fathers, and 20.7% (95% CI 18.4 to 23.0) of other adult caregivers provided high levels of stimulation. Stimulation varied by region, country income group, and Human Development Index (HDI), with higher levels of maternal and paternal–but not other caregivers'–stimulation in high-income and high-HDI countries. Within countries, stimulation levels were, on average, lower in the poorest relative to the richest households, and some but not all countries exhibited differences by child sex (i.e., boys vs. girls) or area (i.e., urban vs. rural).

**Data Availability Statement:** All data for the present study were taken from publicly available data sources. In particular, all MICS files are available from UNICEF's online database at http://mics.unicef.org/surveys. All DHS files are available

from the DHS online database at http://www.
dhsprogram.com/Data/. Country-level data are
available from the World Bank online database at
https://data.worldbank.org/. The authors did not
have any special access privileges to the data.
Interested researchers can replicate the study
findings by directly obtaining the data through the
links provided.

**Funding:** The author(s) received no specific
funding for this work.

**Competing interests:** The authors have declared
that no competing interests exist.

## Conclusions

Results suggest a need for intervention efforts that focus on increasing caregiver stimulation
in LMICs, particularly for fathers and in low-income contexts.

## Introduction

An estimated 250 million children under five are at risk of not achieving their developmental
potential due to inadequate nurturing care. [1, 2] Nurturing care refers to a stable environment
that supports multiple aspects of early childhood development (ECD), including children's
health and nutritional needs, safety and security, opportunities for early learning, and respon-
sive caregiving. [1] Positive and developmentally stimulating environments are especially
important during the earliest years of life, when children's brains develop most rapidly and
responsively to their environments. [3] These early experiences have been found to predict not
only young children's cognitive, language, and socioemotional development, but also their lon-
ger term economic, educational, physical and mental health outcomes. [4, 5]

Caregivers' engagement in stimulation, or play and early learning activities, has been
highlighted as a crucial aspect of nurturing care for supporting children's cognitive and socioe-
motional development [6, 7] Multiple theories of human development–e.g., attachment the-
ory, [8] the bioecological model of human development [9] and relational developmental
systems theory [10]–have long emphasized the importance of caregivers-child interactions as a
critical proximal process for supporting children's development. [8–13] Given that young chil-
dren around the world spend most of their time at home with their main caregivers (i.e., moth-
ers, fathers, grandparents, and others), caregiver-child interactions are even more influential
because these interactions occur on a regular basis over extended periods of time in the imme-
diate environment (i.e., the home) during this developmental period. [9]

Parental stimulation has been linked with a number of cognitive processes in children,
including language [14] and executive function, [15] as well as socioemotional processes such
as persistence and motivation [16] and prosocial behavior. [17] For example, stimulation activ-
ities like book sharing, storytelling or naming or counting objects help children's early lan-
guage and numeracy development. Other activities like playing or taking children outside of
their homes provide children with opportunities to engage in interpersonal interactions, sup-
porting socioemotional skills like prosocial behaviors and emotion regulation. [18] Existing
evidence from economically and culturally diverse countries is consistent with the idea that
stimulation is an important driver of child development, showing strong links between care-
givers' engagement in activities like reading, storytelling, singing songs, venturing outside of
the home, playing, and naming, counting, or drawing objects and children's cognitive and
socioemotional development. [19–22]

Past research has shown both within- and between-country socioeconomic gradients in
maternal stimulation. [7, 18, 23, 24] Families living in low-income contexts are more likely
than their advantaged peers to experience a host of risk factors, including poor health, inade-
quate nutrition, lower levels of parental education, psychosocial stressors, and inadequate ser-
vices, each of which can constrain aspects of nurturing care such as parental stimulation. [25,
26] Moreover, low-income households generally have fewer resources and materials for care-
givers to use with their children for play and early learning (e.g., children's books, toys, house-
hold objects for play). [27]

There are three important limitations in the existing global literature quantifying caregivers' stimulation practices in LMICs. First, the majority of studies have focused exclusively on mothers, [24, 28] despite evidence underscoring that fathers' and other adult caregivers' stimulation may foster young children's positive development, above and beyond mothers' stimulation. [29] Second, little is known about disparities in maternal, paternal, and other caregivers' stimulation related to socioeconomic and demographic characteristics, despite some studies documenting gaps in maternal stimulation [30]. Finally, prior studies on stimulation practices in LMICs are based on older data collected before 2010. [18, 23, 28] Since then, data on caregivers' stimulation practices have been made available in the Multiple Indicator Cluster Surveys (MICS) rounds 4, 5, and 6, and also recently in several rounds of the Demographic and Health Surveys (DHS).

In this paper, we use and combine the latest nationally representative data from the MICS and DHS on caregiver stimulation from 62 LMICs. We estimate the national and total (i.e., average for 62 countries) prevalence of stimulation that mothers, fathers, and other adult caregivers provide to children younger than 5 years in LMICs. We also explore disparities in different caregivers' stimulation based on country-level wellbeing, as measured by the Human Development Index (HDI), and individual and household level sociodemographic characteristics, including household wealth, area of residence, and sex of the child. Specifically, we aim to address three primary research questions:

1. Across 62 economically and culturally diverse countries, what percentage of a) mothers, b) fathers, and c) other adult caregivers provide high levels of home-based stimulation to their young children?

2. How large are the disparities in different caregivers' stimulation between countries based on a) region, b) country income, and c) country-level wellbeing, as measured by the HDI?

3. How large are the disparities in different caregivers' stimulation within countries based on a) household wealth, b) child gender, and c) urbanicity?

## Methods

We used data from 205,150 3- to- 4-year old children and their mothers in 62 LMICs. These data combined 54 MICS (rounds 4–6) and 8 DHS country surveys collected between 2010 and 2018. The MICS and DHS are international household surveys aimed at monitoring the population, health, and wellbeing of women and children younger than five in LMICs. [31] Both surveys employ comparable sampling methodologies to ensure representativeness at the country level, using probabilistic, random samples of households typically drawn from national censuses. S1 Table in the supplementary materials presents a description of the countries included in the study and the households within each country.

The MICS and DHS surveys asked mothers to report whether they (the mother), the child's father, or another household member older than 15 years engaged in the following activities with the child (or a randomly selected child when there was more than one child) in the three days preceding the survey: (1) reading books or looking at picture books; (2) telling stories; (3) singing songs or lullabies; (4) taking the child outside the home; (5) playing with the child; and (6) naming, counting, or drawing things for or with the child. (Importantly, both surveys included the same set of questions about stimulation). These activities have been found to show adequate predictive validity, [19, 22] and to be correlated with children's development and household characteristics as measured by the MICS. [23] Following UNICEF [32] and

previous research conducted with the MICS, [30] we created a count index for the number of activities each caregiver engaged in with the child and defined high stimulation as engagement in least four out of the six activities.

The MICS and DHS also collected information about children's sex, residential area (urban or rural; this information is not available for Argentina), and household wealth quintiles computed using durable asset ownership and access to basic services such as water and sanitation. [33] We also combined data from the UNDP's HDI for the year when each country's survey was conducted to characterize country-level wellbeing. The HDI is a composite measure of life expectancy, education, and per capita income, [34] and has been found to correlate with maternal engagement in cognitive and socioemotional stimulation [7] and other parental practices [35] in LMICs. Additional country-level data included: Gini index, [36] percentage of urban population, and unemployment rate [37], and country region and income group as determined by the World Bank. [38]

To address RQ1, we estimated the percentage of mothers, fathers, and other caregivers who engaged in high stimulation and 95% confidence intervals around these estimates using MICS- and DHS-provided sampling weights to ensure national representativeness. We also estimated caregivers' engagement in each of the six activities separately. Using this information, we addressed RQ2 by estimating the percentage of mothers, fathers, and other caregivers who engaged in high stimulation in six regions (i.e., East Asia & Pacific; Europe & Central Asia; Latin America & the Caribbean; Middle East & North Africa; South Asia; Sub-Saharan Africa) and three income groups (low-income; lower-middle income; upper-middle income), as well as 95% confidence intervals around these estimates. Moreover, we assessed between-country disparities in the percentage of children exposed to high stimulation according to the HDI both descriptively and using a multivariate regression model adjusting for country-level income inequality, urbanicity, unemployment, country region and income group. To addressed RQ3, we estimated within-country absolute differences in the percentage of children exposed to high stimulation by each caregiver, by wealth quintile (richest vs. poorest), sex (boys vs. girls), and residential area (urban vs. rural), including 95% confidence intervals around these estimates to assess statistically significant disparities. We conducted all analyses using Stata 16.0. [39]

## Results

Table 1 presents the aggregate proportion of children exposed to high stimulation by their mothers, fathers, and other caregivers in 62 available LMICs. As shown in S2 Table, sampled countries are not different, on average, from countries excluded from the study in key characteristics. On average, 39.8% of children were exposed to high stimulation from their mothers, 11.9% from their fathers, and 20.7% from other caregivers older than 15 years. On average, high levels of maternal stimulation were most common in Europe and Central Asia (70.5%) and least common in Sub-Saharan Africa (14.6%). High levels of paternal stimulation were most common in Europe and Central Asia (21%) and least common in Sub-Saharan Africa (3.9%), whereas high levels of other caregivers' stimulation were most common in Latin America and the Caribbean (23.2%) and least common in Middle East and North Africa (15.4%). Moreover, both maternal and paternal stimulation exhibited an income gradient, where low-income countries had the lowest proportion of children exposed to high maternal and paternal stimulation and upper-middle income countries had the highest proportions, whereas no gradient was revealed for other caregivers' high stimulation.

Fig 1 shows a positive bivariate correlation between country-level HDI and the proportion of children exposed to high maternal ($r = 0.84$; $p < 0.001$) and paternal ($r = 0.72$; $p < 0.001$)

**Table 1. Proportion of children exposed to high stimulation by caregiver and 95% CI.**

| | Maternal | Paternal | Other caregivers |
|---|---|---|---|
| Region | | | |
| East Asia & Pacific | 35.6 (33.4, 37.8) | 15.1 (13.4, 16.7) | 22.6 (20.7, 24.5) |
| Europe & Central Asia | 70.5 (67.5, 73.4) | 21.0 (18.1, 24.0) | 22.1 (19.0, 25.2) |
| Latin America & the Caribbean | 55.8 (51.8, 59.8) | 16.0 (12.8, 19.2) | 23.2 (19.6, 26.8) |
| Middle East & North Africa | 48.0 (45.5, 50.5) | 14.9 (13.1, 16.8) | 15.4 (13.5, 17.2) |
| South Asia | 40.7 (39.0, 42.4) | 11.9 (10.2, 13.5) | 22.6 (20.6, 24.6) |
| Sub-Saharan Africa | 14.6 (13.2, 15.9) | 3.9 (3.2, 4.6) | 19.2 (17.7, 20.7) |
| Income group | | | |
| Low-income | 17.4 (16.2, 18.6) | 4.8 (4.1, 5.5) | 19.2 (17.9, 20.5) |
| Lower-middle income | 31.8 (29.5, 34.1) | 8.5 (7.2, 9.9) | 19.3 (17.3, 21.3) |
| Upper-middle income | 63.1 (59.7, 66.6) | 20.2 (17.0, 23.4) | 23.0 (19.7, 26.3) |
| Average for 62 countries | 39.8 (37.4, 42.2) | 11.9 (10.1, 13.8) | 20.7 (18.4, 23.0) |

95% CI in parentheses.

stimulation, and a positive but not statistically significant correlation between the HDI and children's exposure to other caregivers' stimulation ($r = 0.16$; $p = 0.21$). As shown in S3 Table, the statistically significant country-level association between the HDI and the proportion of children exposed to high maternal ($\beta = 1.34$; $p < 0.01$) and paternal stimulation ($\beta = 0.47$; $p < 0.1$) holds even after accounting for other country-level characteristics such as income inequality, urbanicity, unemployment, region, and income group. Similarly, no significant association was found between the HDI and other caregivers' high stimulation when adjusting for these covariates ($\beta = 0.39$; $p > 0.1$).

Fig 2 shows the country-level percentages of children exposed to high levels of stimulation by their mothers, fathers, and other caregivers (see S4 Table for additional details). The countries with the highest proportions of high maternal stimulation were Montenegro (91.9%), Serbia (89.6%) and the Maldives (86.89), whereas the countries with the lowest proportions were the Democratic Republic of the Congo (4.7%), Afghanistan (4.5%), and Guinea-Bissau (2.9%). The proportions of children exposed to high paternal stimulation were systematically lower than those exposed to high maternal stimulation in all the countries included in the analysis, even in the countries with the highest paternal stimulation, which were Montenegro (45.5%), Serbia (36.6%) and Thailand (34.4%), and in those where paternal stimulation was the lowest, such as The Gambia (0.8%), Senegal (0.8%), and Guinea-Bissau (0.3%). Finally, high levels of other caregivers' engagement in stimulating activities were most prevalent in Thailand (52.6%), St. Lucia (41.1%), and Uruguay (38.2%), and least prevalent in Benin (6.4%), Timor-Leste (4.5%), and Sierra Leone (3.1%). Importantly, neither paternal nor other caregivers' high stimulation had a prevalence greater than 60% in any of the included countries.

Further analysis of each individual activity showed that other caregivers were more engaged in activities such as playing or going out with the child in comparison to fathers and even mothers, particularly in Sub-Saharan Africa and South Asia (see S5 Table for details). For example, in 10 Sub-Saharan countries a higher proportion of other caregivers engaged in five or more individual activities relative to mothers and fathers. Similarly, in 28 out of 62 countries included in the study, other caregivers' engagement was higher than that observed for fathers across all individual assessed activities.

Fig 3 presents country-level wealth disparities in the proportion of children exposed to high levels of stimulation by different caregivers (see S6–S8 Tables for details). Higher proportions

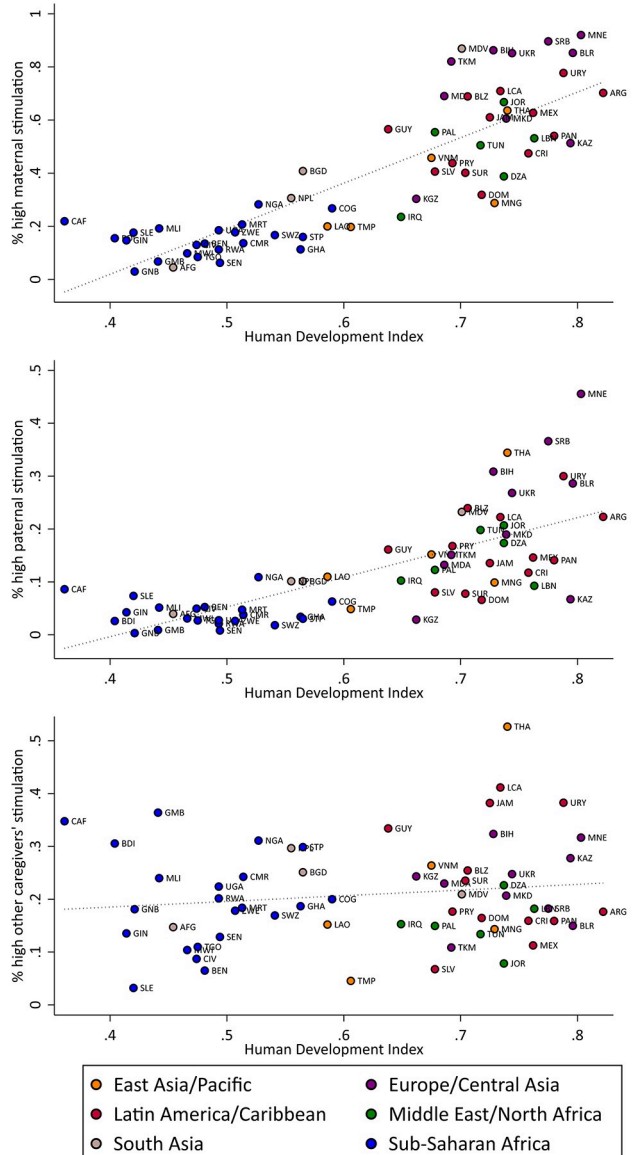

**Fig 1. Proportion of children exposed to high stimulation by caregiver by HDI.**

of high maternal and paternal stimulation were systematically found in the richest households (i.e., fifth quintile) in comparison to the poorest households (i.e., first quintile). In 57 and 48 out of 62 countries high levels of maternal and paternal stimulation, respectively, were statistically significantly more common in the richest households than in the poorest (See S6 and S7 Tables). In addition, in 27 countries the proportion of high stimulation by other caregivers was higher in the richest than the poorest households, but in three countries (Central African Republic, Kazakhstan, and Togo) there were higher levels of other caregivers' engagement in stimulating activities in the poorest than in the richest households (S8 Table). Fig 4 presents the proportion of stimulation disaggregated by child sex. In 10 countries high maternal and paternal stimulation was significantly more prevalent for boys than for girls, whereas this child gender difference was only observed in one country (Guinea-Bissau) for other caregivers'

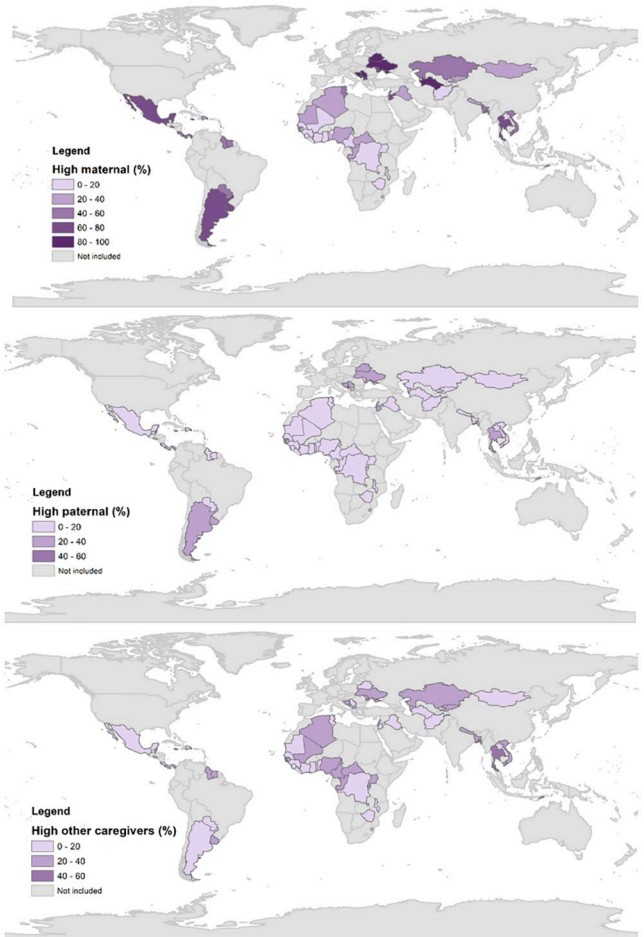

**Fig 2. Proportion of children exposed to high stimulation by caregiver.** See S4 Table for additional details.

stimulation (see S9–S11 Tables). Fig 5 presents disparities in stimulation by area of residence. High levels of maternal, paternal, and other caregivers' stimulation were significantly more common in urban than rural areas in 44, 34, and 15 countries, respectively, whereas the opposite trend was observed in 1 country for paternal stimulation and 10 countries for other caregivers' stimulation (S12–S14 Tables).

## Discussion

In this study, we used data from 62 LMICs to estimate the proportion of mothers, fathers, and other adult caregivers who provide high levels of stimulation to children aged 3- and 4- years. Our results revealed that, on average, 39.8% of mothers and 11.9% of fathers engage in high levels of stimulation (i.e., at least 4 of 6 activities in the past 3 days) in LMICs. These findings are consistent with prior research [18, 29] that have highlighted significantly lower levels of paternal than maternal stimulation in LMICs and are not surprising considering the predominantly patriarchal norms and expectations regarding women's household and caregiving responsibilities globally. [40]

We found substantial wealth-related variability across and within countries in the percentage of mothers and fathers' stimulation, confirming results from previous studies. [7, 23, 30] Countries with low levels of HDI and the poorest households within countries had, on average,

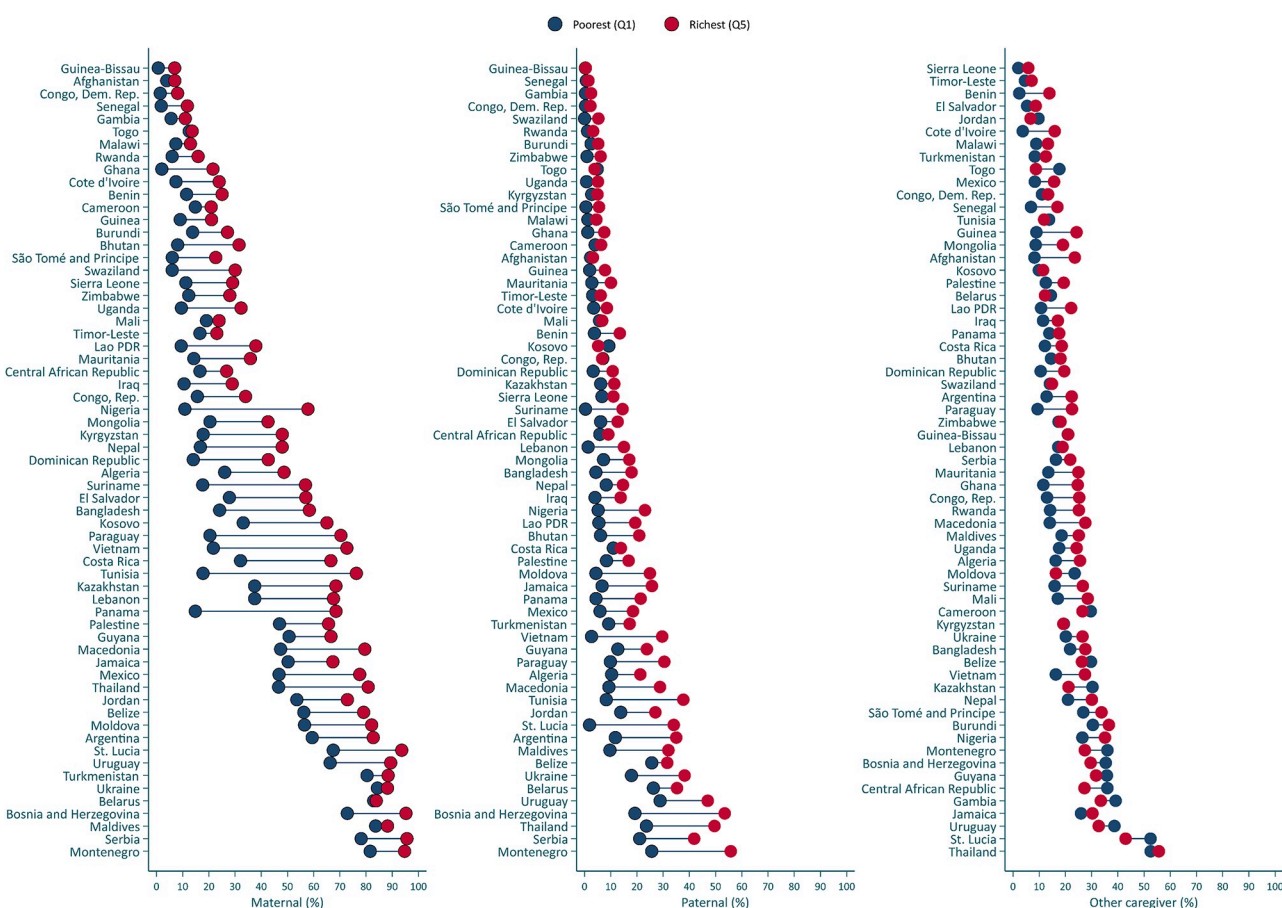

**Fig 3. Proportion of children exposed to high stimulation by caregiver by wealth quintile.** See S6–S8 Tables for additional details.

lower levels of maternal and paternal stimulation relative to better-off countries and households. One potential mechanism underlying this association is that in low-income countries and households a higher proportion of parents who are illiterate or have low levels of education, [41] have more children to take care of, [42] or experience poverty. All these factors may constrain parental caregiving capacities or shift parents' priorities for their young children (e.g., focus more on illness prevention and health care seeking than stimulation). [26, 43] For example, low levels of education and income poverty have been found to compromise parents' capacity to engage in stimulating activities with their young children through exacerbating contextual stressors [26, 44] Similarly, prior studies show that having more children to take care of reduces caregivers' capacity to provide high levels of stimulation to any one child. [45] Even though these explanations are plausible, future studies should examine the specific mechanisms explaining the disparities that we document in this study.

We also found that a substantial proportion of other caregivers (20.7%) engage in high levels of stimulation in LMICs. These results align with a growing body of evidence that highlights the active childrearing roles of grandparents, older siblings, and other caregivers in LMICs. [46–49] At the same time we found that the HDI did not correlate with other caregivers' provision of high levels of stimulation and that wealth-related disparities were less salient for other caregivers' stimulation compared to maternal or paternal stimulation. These findings could

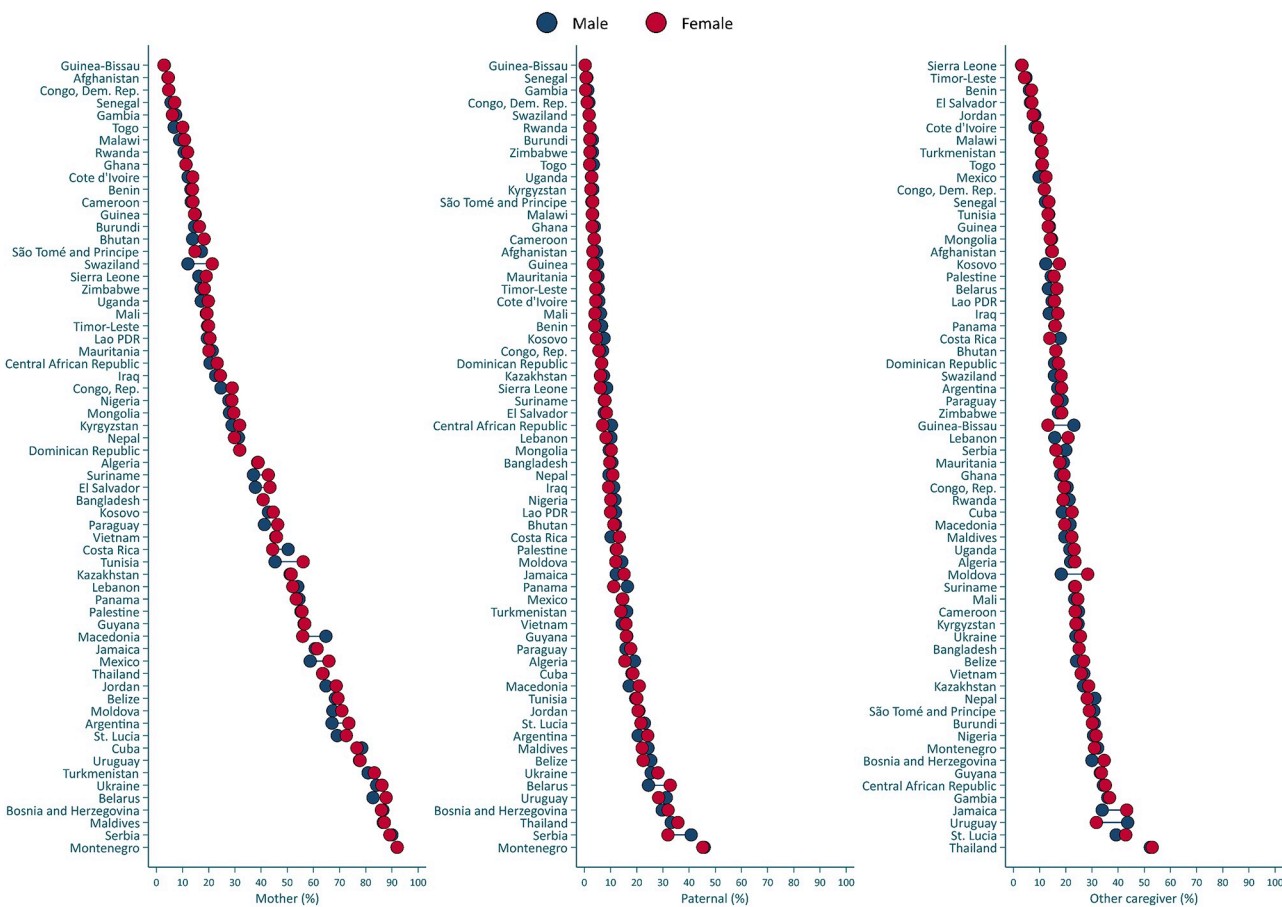

**Fig 4. Proportion of children exposed to high stimulation by caregiver by child sex.** See S9–S11 Tables for additional details.

indicate differences in cultural norms about caregiving above and beyond contextual (resource) constraints. [49, 50]

Furthermore, we observed substantial differences in maternal, paternal, and other caregivers' engagement in each stimulation activity assessed. Interestingly, other caregivers (e.g., grandparents, siblings), in addition mothers and fathers, engage in high levels of stimulation in different individual activities in multiple Sub-Saharan African and South Asian countries. These findings echo ethnographic studies suggesting that the exclusive focus on the mother-child dyad in research conducted with Western samples may be overly restrictive in non-Western settings where childrearing is a shared responsibility among other household and community members. [49] The fact that other caregivers' engagement in Sub-Saharan Africa and South Asia is particularly pronounced for some activities (e.g., playing) may further underscore the relevance of cultural differences in caregiving across settings.

Finally, our study revealed inequalities in different caregivers' stimulation across urban and rural areas, which could also be related to the overall higher prevalence of poverty in rural relative to urban areas in LMICs. [51] In contrast, we did not find consistent evidence to support that caregivers' stimulation differed by child sex in the 62 countries included in the study. These results align with prior studies that have identified the same factors as sources of variation for stimulation [23, 28] and early child education. [24]

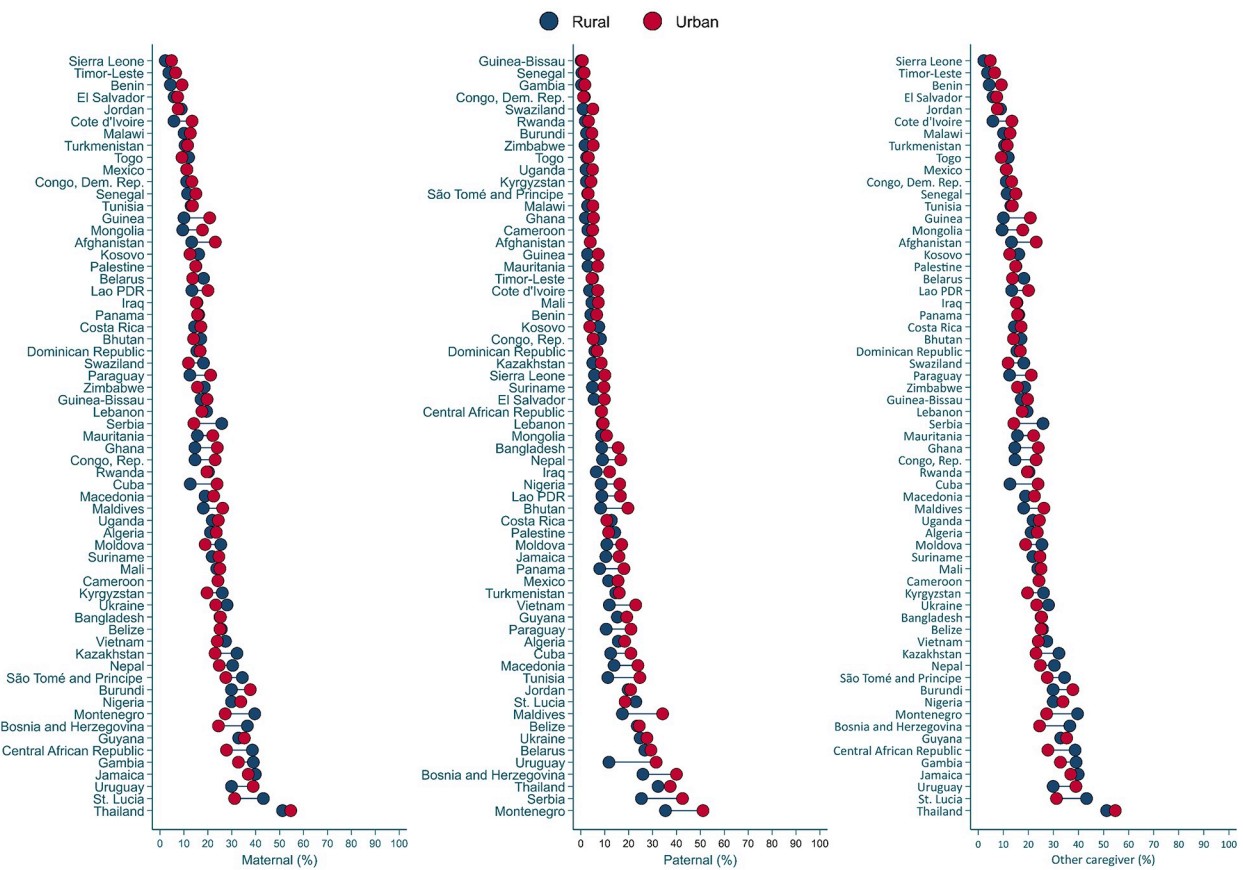

**Fig 5. Proportion of children exposed to high stimulation by caregiver by area of residence.** See S12–S14 Tables for additional details.

This study has important limitations that should be discussed. First, the measures of caregivers' stimulation are based on maternal reports exclusively, which may potentially bias our estimates. For example, some mothers may underreport paternal or another caregiver's engagement in stimulation if they did not know about the interaction or overreport for social desirability. Second, there are likely to be other important stimulation activities specific to cultural groups that were not assessed in the current study. [50] In this sense, these results may under-represent the true levels of stimulation that children are exposed to in diverse parts of the world. Third, the dataset used in the study did not allow us to characterize the frequency or quality of the stimulating activities different caregivers provided to the children. More research is warranted to determine what "high-quality" stimulation means within different cultural contexts with varying social norms and parenting goals and attitudes, [49, 52] and to understand sources of variation in such quality. Fourth, the current analyses did not allow us to understand the sources (or mechanisms) of wealth- and- area-related variation in caregivers' stimulation within and between countries, such as differences in household sizes or cultural norms around childrearing. Finally, the current study focused on children aged 36–59 months, so findings cannot generalize to caregivers' engagement in stimulating activities with younger or older children.

Despite these limitations, the present study contributes to a nascent body of literature examining the role of different caregivers' practices in non-western or industrialized settings.

[49, 53] Considering the associations between caregivers' stimulation and young children's development identified in previous studies, [29, 54] our findings underscore the opportunity and potential for supporting not only maternal but also paternal and other caregivers' stimulation in LMICs. Stimulation interventions have been implemented effectively at scale with mothers in diverse cultural settings (e.g., Colombia, Jamaica, Pakistan), demonstrating positive impacts on maternal practices and ECD. [55] An emerging body of research in LMICs has also demonstrated the effectiveness of father involvement interventions for promoting paternal stimulation and early child development outcomes. For example, a parenting intervention in Vietnam encouraged fathers to responsively interact with their infants and work together with the mother as part of a parenting team. Results revealed that the intervention improved father-child relationships and infants' language, socioemotional, and motor development outcomes. [56] Although fewer programs have intentionally engaged other caregivers in parenting programs, a pilot trial of the Triple P program was conducted with grandparents of preschoolers in Hong Kong and found reductions in grandchildren's behavior problems. [57]

Overall, our study findings suggest that parenting programs such as these are critically important for caregivers and young children globally. Future programs should engage not only mothers, but also other caregivers from a family-inclusive perspective, to enhance relationships between children and multiple caregivers and potentially increase the effectiveness of intervention strategies to improve children's early cognitive and socioemotional development. Policies to promote caregiver stimulation are also needed to ensure maximal reach of such programs at scale.

## Conclusion

The present study reveals substantial variability in different caregivers' engagement in stimulating activities with young children in LMICs, with overall lower stimulation from fathers relative to mothers and other caregivers. The study also highlights considerable disparities in different caregivers' stimulation between and within countries. More research is needed to identify contextual factors that may impede or promote caregivers' engagement in their stimulating activities with children. Doing so will contribute to designing effective programs to support multiple caregivers and advancing our understanding of the role of cultural factors in determining how caregivers' stimulation manifests in global settings, with the ultimate goal of promoting child development globally.

## Supporting information

**S1 Table. Sample characteristics.**
(DOCX)

**S2 Table. Included vs. excluded LMICs characteristics.**
(DOCX)

**S3 Table. Association between the proportion of children exposed to high stimulation and country-level characteristics for 62 countries.**
(DOCX)

**S4 Table. Percentage of children exposed to high stimulation (four of more activities).**
(DOCX)

**S5 Table. Prevalence of maternal, paternal and other caregivers' stimulation by activity (percentage of people engaged in each activity).**
(DOCX)

**S6 Table. Wealth disparities in the percentage of children exposed to high maternal stimulation.**
(DOCX)

**S7 Table. Wealth disparities in the percentage of children exposed to high paternal stimulation.**
(DOCX)

**S8 Table. Wealth disparities in the percentage of children exposed to high stimulation by other caregivers.**
(DOCX)

**S9 Table. Sex disparities in the percentage of children exposed to high maternal stimulation.**
(DOCX)

**S10 Table. Sex disparities in the percentage of children exposed to high paternal stimulation.**
(DOCX)

**S11 Table. Sex disparities in the percentage of children exposed to high stimulation by other caregivers.**
(DOCX)

**S12 Table. Area disparities in the percentage of children exposed to high maternal stimulation.**
(DOCX)

**S13 Table. Area disparities in the percentage of children exposed to high paternal stimulation.**
(DOCX)

**S14 Table. Area disparities in the percentage of children exposed to high stimulation by other caregivers.**
(DOCX)

## Author Contributions

**Conceptualization:** Jorge Cuartas, Joshua Jeong, Catalina Rey-Guerra, Dana Charles McCoy, Hirokazu Yoshikawa.

**Data curation:** Jorge Cuartas.

**Formal analysis:** Jorge Cuartas, Joshua Jeong, Catalina Rey-Guerra.

**Investigation:** Jorge Cuartas, Joshua Jeong, Catalina Rey-Guerra, Dana Charles McCoy, Hirokazu Yoshikawa.

**Methodology:** Jorge Cuartas, Joshua Jeong, Catalina Rey-Guerra, Dana Charles McCoy, Hirokazu Yoshikawa.

**Project administration:** Jorge Cuartas.

**Software:** Jorge Cuartas, Catalina Rey-Guerra.

**Validation:** Jorge Cuartas, Joshua Jeong, Catalina Rey-Guerra, Dana Charles McCoy, Hirokazu Yoshikawa.

**Visualization:** Jorge Cuartas.

**Writing – original draft:** Jorge Cuartas, Joshua Jeong, Catalina Rey-Guerra.

**Writing – review & editing:** Jorge Cuartas, Joshua Jeong, Catalina Rey-Guerra, Dana Charles McCoy, Hirokazu Yoshikawa.

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
