## [Decision Letter · Decision Letter 0]

22 Apr 2020

PONE-D-20-04986

Caregivers’ Cognitive and Socioemotional Stimulation in Low- and- Middle-Income Countries

PLOS ONE

Dear Mr. Cuartas,

Thank you for submitting your manuscript to PLOS ONE. After careful consideration, we feel that it has merit but does not fully meet PLOS ONE’s publication criteria as it currently stands. Therefore, we invite you to submit a revised version of the manuscript that addresses the points raised during the review process.

We would appreciate receiving your revised manuscript by Jun 06 2020 11:59PM. To enhance the reproducibility of your results, we recommend that if applicable you deposit your laboratory protocols in protocols.io, where a protocol can be assigned its own identifier (DOI) such that it can be cited independently in the future. For instructions see: http://journals.plos.org/plosone/s/submission-guidelines#loc-laboratory-protocols

We look forward to receiving your revised manuscript.

Kind regards,

Thach Duc Tran, M.Sc., Ph.D.

Academic Editor

PLOS ONE

Journal Requirements:

2. Please ensure that you include a title page within your main document.

We do appreciate that you have a title page document uploaded as a separate file, however, as per our author guidelines (http://journals.plos.org/plosone/s/submission-guidelines#loc-title-page) we do require this to be part of the manuscript file itself and not uploaded separately.

3. Please also amend your manuscript to include your abstract after the title page.

Reviewers' comments:

Reviewer's Responses to Questions

**Comments to the Author**

1. Is the manuscript technically sound, and do the data support the conclusions?

Reviewer #1: Yes

Reviewer #2: Yes

2. Has the statistical analysis been performed appropriately and rigorously? 

Reviewer #1: Yes

Reviewer #2: Yes

3. Have the authors made all data underlying the findings in their manuscript fully available?

Reviewer #1: Yes

Reviewer #2: Yes

4. Is the manuscript presented in an intelligible fashion and written in standard English?

Reviewer #1: Yes

Reviewer #2: Yes

5. Review Comments to the Author

Reviewer #1: I congratulate the authors because this is a most needed area of study. I hope you will continue studying MICS and other demographic data. The following are three comments to your paper. I leave it up to you to consider them or not:

1) The socioemotional piece is missing in the Introduction (although is in the title). In future publications it would be important to define it properly explaining how it can be affected by the 6 MICS / DHS measures used in the study.

2) In future publications consider using Hierarchical Linear Modeling (this regression technique is important when groups within groups are studied).

3) Consider adding a few lines to you Conclusion. What you found has implications in the need of Parenting Programs worldwide. Parenting programs allow to bring children and caregivers together and work on socioemocional and cognitive stimulation. Please sell your findings!!!!

Reviewer #2: This is a well-written manuscript addressing the caregivers’ cognitive and socioemotional stimulation in low- and middle-income countries using publicly available datasets from MICS and DHS. I have following suggestions for the author to further revise this manuscript before considering its publication in PLOS One.

1. The theoretical framework is not clear to support the work reported in this manuscript. Relevant work conducted in prior studies can be reviewed in a more systematic way. The research questions can be more specific and explicit.

2. More details of the Methods should be provided, including the source of the data (how many from MICS and how many from DHS, how to combine the datasets, which variable from which source), and the statistical methods used to address each of the research questions.

3. The discussion is general currently. Potential mechanism related to the findings for each research questions can be further discussed.

6. PLOS authors have the option to publish the peer review history of their article (what does this mean?). If published, this will include your full peer review and any attached files.

Reviewer #1: Yes: Alfredo R. Tinajero, PhD

Reviewer #2: No

---

## [Author Response · Author response to Decision Letter 0]

19 May 2020

Editor

Thank you very much for the opportunity to revise the manuscript. The Reviewers’ comments were very helpful, and we made several edits to our manuscript in response to each comment, as we describe below. 

Reviewer #1 

1. I congratulate the authors because this is a most needed area of study. I hope you will continue studying MICS and other demographic data. The following are three comments to your paper. I leave it up to you to consider them or not

We thank the Reviewer for the review and helpful feedback. We have made edits throughout the manuscript following the Reviewer’s suggestions.

2. The socioemotional piece is missing in the Introduction (although is in the title). In future publications it would be important to define it properly explaining how it can be affected by the 6 MICS / DHS measures used in the study.

We agree with the Reviewer and have decided to expand our discussion on the importance of caregivers’ stimulation for both cognitive and socioemotional development in the specific context of low- and- middle-income countries in the Introduction. In particular, we provide expanded discussion of the type of socioemotional skills impacted by stimulation, with examples on pages 2-3 of how different activities captured in the MICS / DHS can promote cognitive and socioemotional skills. Moreover, we now mention empirical findings showing strong links between the six MICS / DHS activities assessed by the stimulation measure in our study and children’s cognitive and socioemotional development. 

3. In future publications consider using Hierarchical Linear Modeling (this regression technique is important when groups within groups are studied).

We thank the Reviewer for this suggestion. The current paper is merely descriptive and does not assess any statistical predictive association at the individual level or between a level-2 and level-1 variable as might be done in HLM (that is, associations calculated as correlations are solely at the country level, such as the association between HDI and proportion of children exposed to different levels of stimulation). To ensure that our country-level samples were each nationally representative, we employed country-specific sample weights, as is suggested when using the MICS and DHS. We agree that future studies with the MICS or DHS examining associations between the stimulation index and child outcomes or other variables could benefit from employing hierarchical linear models to account for the nesting of children and families within countries. 

4. Consider adding a few lines to you Conclusion. What you found has implications in the need of Parenting Programs worldwide. Parenting programs allow to bring children and caregivers together and work on socioemocional and cognitive stimulation. Please sell your findings!!!

We followed the Reviewer’s suggestion and included an additional paragraph at the end of the discussion section reviewing some programs that have tried to bring different caregivers together and discussing the importance of such kind of programs in LMICs. Please see pages 14-15.

Reviewer #2

5. This is a well-written manuscript addressing the caregivers’ cognitive and socioemotional stimulation in low- and middle-income countries using publicly available datasets from MICS and DHS. I have following suggestions for the author to further revise this manuscript before considering its publication in PLOS One.

We thank the Reviewer for the review and useful comment. We have made edits throughout the manuscript following the Reviewer’s suggestions.

6. The theoretical framework is not clear to support the work reported in this manuscript. Relevant work conducted in prior studies can be reviewed in a more systematic way. The research questions can be more specific and explicit.

We thank the Reviewer for this important comment. We have now added more details to the introduction of the manuscript with regard to the theoretical framework. In particular, we included an additional paragraph discussing prior theories showing the importance of caregivers’ stimulation for young children’s development, including the bioecological model of development; relational developmental systems theory, and attachment theory. We have also included additional citations of prior work on pages 3-4. Finally, we have clarified the three research questions that we address in the study (see p. 5) following the Reviewer’s suggestion. 

7. More details of the Methods should be provided, including the source of the data (how many from MICS and how many from DHS, how to combine the datasets, which variable from which source), and the statistical methods used to address each of the research questions.

We have made edits in the manuscript and supplementary materials in response to this comment. First, we mention in the manuscript the number of countries that come from the MICS and from the DHS, as follows: “These data combined 54 MICS (rounds 4-6) and 8 DHS country surveys collected between 2010 and 2018.” Furthermore, we present details on each country’s sample characteristics in S1 Table (supplementary materials). Finally, we edited the methods section to (1) include more details on the statistical methods and (2) link each research question with the statistical methods used, as suggested by the Reviewer (see p. 7). Moreover, we clarified that both the MICS and DHS included the same set of questions about stimulation (see p. 6), in response to the Reviewers’ comment.

8. The discussion is general currently. Potential mechanism related to the findings for each research questions can be further discussed.

We thank the Reviewer for this comment. We included additional details in the discussion speculating about potential mechanisms explaining our findings. We also mention that future studies should examine in detail the mechanisms underlying our results. Please see pages 11-14.

---

## [Editor Report · Decision Letter 1]

30 Jun 2020

Maternal, Paternal, and other Caregivers’ Stimulation in Low- and- Middle-Income Countries

PONE-D-20-04986R1

Dear Dr. Cuartas,

We’re pleased to inform you that your manuscript has been judged scientifically suitable for publication and will be formally accepted for publication once it meets all outstanding technical requirements.

Kind regards,

Thach Duc Tran, M.Sc., Ph.D.

Academic Editor

PLOS ONE
---

## [Editor Report · Acceptance letter]

1 Jul 2020

PONE-D-20-04986R1 

Maternal, Paternal, and other Caregivers’ Stimulation in Low- and- Middle-Income Countries 

Dear Dr. Cuartas:

I'm pleased to inform you that your manuscript has been deemed suitable for publication in PLOS ONE. Congratulations! Your manuscript is now with our production department. 

Kind regards, 

on behalf of

Dr. Thach Duc Tran 

Academic Editor

PLOS ONE